# Linear feedback coding scheme for multiple-access fading channels with degraded message sets

Yuan Liao, Xiaofang Wang *

School of Intelligence Technology, Geely University of China, Chengdu, Sichuan, China

☯ These authors contributed equally to this work.
* wangxiaofang@guc.edu.cn

**Data Availability Statement:** All relevant data are within the paper.

**Funding:** The authors received no specific funding for this work.

## Abstract

Channel coding technology plays an important role in wireless communication systems, and it serves as a crucial mechanism to reduce interference during the transmission process. As the fifth-generation (5G) and sixth-generation (6G) wireless communication systems rapidly advance, requirements of the users on the quality and security of wireless service are increasing. To solve these problems, it calls for us to explore the new channel coding technologies. In this paper, a linear feedback coding scheme for fading multiple-access channels with degraded message sets (FMAC-DMS) is proposed. In this scheme, the transmitting beamforming and channel splitting are used to transform the channel with complex signals into scalar equivalent sub-channels. Then, the extended Schalkwijk-Kailath coding scheme (SK) is further applied to each sub-channel. The channel capacity, finite blocklength (FBL) sum-rate and FBL secrecy achievable sum-rate of the FMAC-DMS in single-input single-output (SISO) and multi-input single-output (MISO) cases are derived. Finally, we show that the proposed scheme not only provides a FBL coding solution but also guarantees physical layer security(PLS). The numerical and simulation results show the effectiveness of the proposed scheme as a channel coding solution. The study of this paper provides a new method to construct a practical FBL scheme for the FMAC-DMS.

## 1 Introduction

The fastest-growing field in the telecommunications industry is wireless communication systems, with 5G currently in the early stages of commercial deployment worldwide. However, researchers need to develop new enabling technologies to overcome existing technical barriers. Channel coding, as one of the fundamental techniques in wireless communication systems, plays a crucial role in error detection and correction. It is the primary method to combat noise interference and improve signal quality during transmission. To meet the growing demands for high reliability and low latency, exploring new channel coding schemes is necessary.

Multiple-access fading channel (FMAC) is a typical channel model in wireless communication, and researchers [1–5] have studied linear feedback coding schemes for FMAC from an information theory perspective. Among them, [1–3] have extended the classical Schalkwijk-

**Competing interests:** The authors have declared that no competing interests exist.

Kailath (SK) coding scheme [6] to wireless intelligent reflecting surfaces, single-input multi-output (SIMO) Rayleigh channels, and two-way full-duplex fading channels, respectively. These proposed schemes have been shown to possess self-secrecy when the code length exceeds a certain threshold, significantly enhancing the confidentiality of the target communication system and ensuring physical layer security (PLS). In [4], a linear coding scheme for fading broadcast channels with additive Gaussian white noise (AWGN) was proposed, which achieves rates close to the posterior matching scheme derived in [4] by choosing specific parameters.

With the deepening research on channel coding, many researchers have turned their attention to the field of PLS. [7, 8] have indicated that feedback signals can generate a secret key and encrypt the transmitted messages during the transmission process, thereby enhancing the secrecy capacity of discrete memoryless wiretap channels to the same channel capacity as the unconstrained secrecy case. However, these key-based feedback schemes are constructed based on Shannon's classical random coding theory [9] and are not practical or implementable in real communication systems. Therefore, there is a need for research on finite blocklength (FBL) coding schemes. [9] proved that the classical SK feedback coding scheme [6] is an excellent FBL coding scheme for point-to-point Gaussian white noise channels. This paper, based on the demands for high reliability, low latency, and data protection in communication, takes an information-theoretic approach and combines the favorable properties of SK coding to design a linear feedback coding scheme for single-input single-output fading multiple-access channels with degraded message sets (SISO-FMAC-DMS), satisfying the requirements of PLS. The following questions are the focus of this study:

1. Can the classical SK coding scheme be extended to SISO-FMAC-DMS, considering its own favorable properties?

2. Can the proposed scheme be extended to multi-input single-output fading multiple-access channels with degraded message sets (MISO-FMAC-DMS)?

3. Can the proposed scheme ensure physical layer security (PLS) in the presence of eavesdroppers?

In this paper, we first present the main idea of the layered SK scheme and apply it to FMAC-DMS in both single-input single-output (SISO) and multi-input single-output antenna systems (MISO). The channel capacities and FBL achievable rate expressions for both scenarios are derived. The FBL achievable secure rate expressions are also analyzed in the presence of eavesdroppers. Finally, data simulations are performed, and compared to classical LDPC codes, Turbo codes, and LDGM codes, the proposed layered SK coding scheme significantly reduces the blocklength. Further analysis of the characteristics of this scheme is conducted.

This paper is organized as follows. In Section 2, we briefly review the related works. Formal definition of the SISO-FMAC-DMS and MISO-FMAC-DMS models studied in this paper and previous results is introduced in Section 3. Our proposed coding schemes for SISO-F-MAC-DMS and MISO-FMAC-DMS are shown in Section 4. Section 5 shows that our proposed scheme for the MISO-FMAC-DMS is also a secure FBL scheme when the coding blocklength is lager than a certain threshold. Simulation results and numerical examples are given in Section 6. Section 7 summary of all results in this paper and discusses future work.

## 2 Discussion

As early as 1966, Schalkwijk and Kailath [6] proposed a capacity-achieving coding scheme for point-to-point channels with AWGN, known as the SK coding scheme. The SK coding scheme

**Table 1. Summarizing all works in the linear feedback coding scheme.**

| Related Work | siso | simo | PLS | MAC | DMS |
|---|---|---|---|---|---|
| [1] | ✓ | ✓ | ✓ | FMAC | – |
| [2] | – | ✓ | ✓ | – | – |
| [3] | ✓ | – | ✓ | – | – |
| [6] | ✓ | – | – | – | – |
| [10] | ✓ | – | – | ✓ | – |
| [11] | ✓ | – | – | $L$-user ($L \geq 3$) Gaussian MAC | – |
| [12] | ✓ | – | ✓ | Gaussian MAC | ✓ |
| This Work | ✓ | ✓ | ✓ | FMAC | ✓ |

[6] is an excellent constructive coding scheme for the white Gaussian channel, and it has extremely low complexity in encoding and decoding. Furthermore, the decoding error probability of SK scheme *doubly exponentially decays* as the coding code length increases, which indicates that the SK scheme [6] necessitates an extremely short coding code length to achieve a target decoding error probability. Then, Ozarow [10] extended the SK coding [6] to the multiple access channel (MAC) with AWGN, it showed that the feedback can increase the achievable rate region of the MAC, and this coding scheme is also capacity-achieving for the MAC with feedback. In [11], the SK-type scheme proposed in [10] was further extended to $L$-user ($L \geq 3$) Gaussian MAC with noiseless feedback, and the sum-rate capacity was determined for some specific cases. Recently, [1] has extended the classical SK scheme [6] to intelligent reflecting surfaces in the presence of an eavesdropper, and it was shown that the proposed scheme is self-secure when the code length exceeds a certain threshold. Reference [2] extended the classical SK scheme [6] to both SIMO Rayleigh channels and wire-tap channels, and it demonstrated that the extended scheme is self-secure capacity-achieving. In [3], the classical SK scheme [6] was extended to two-way full-duplex Gaussian channels, and further prove that the proposed scheme is self-secure. A self-secure capacity-achieving feedback scheme for the Gaussian MAC with degraded message sets (DMS) is proposed in [12], which is an extended version of the classical SK scheme [6]. Furthermore, a summary of all works in the linear feedback coding scheme can be found in Table 1.

Though the SK-type coding scheme is well investigated in various channel models, however, SK-type FBL scheme for the FMAC with DMS remains unknown. In this paper, we focus on the SISO-FMAC-DMS and MISO-FMAC-DMS models and study how to design the SK-type FBL schemes for these models. Additionally, we also investigated whether these proposed schemes can achieve perfect secrecy by themselves.

## 3 Model formulation

For convenience, Table 2 summarizes the key notations introduced in this paper.

*Assumptions*:(1) In low-latency communication systems, the transmission time of data packets is much shorter compared to the time scale of fading variations. Therefore, such communication channels are often considered as quasi-static fading. Hence, we assume that the forward link and its feedback link are static fading channels, where the channel coefficients remain constant during the transmission process, and both the receiver and the transmitters perfectly know the channel state information (CSI).

(2) From similar argument in [13], we assume that the external eavesdropper is an active user but is untrusted by the legitimate receiver, which indicates that the perfect CSI of

**Table 2. Key notations.**

| Symbols | Definitions |
|---|---|
| $N$ | Codeword blocklength |
| $\varepsilon$ | Decoding error probability |
| $\mathcal{N}(0, \sigma^2)$ | Gaussian distribution with mean 0 and variance $\sigma^2$ |
| $\mathcal{CN}(0, \sigma^2)$ | Circularly symmetric complex Gaussian distribution with mean 0 and variance $\sigma^2$ |
| $\mathbb{C}^{M \times N}$ | A complex matrix of size $M \times N$ |
| $U(a, b)$ | A uniform distribution on the interval $[a, b]$ |
| $\|\cdot\|$ | The 2-norm of a vector |
| $|\cdot|$ | The cardinality of a finite set |
| $\bar{X}$ | The conjugate of the complex-valued $X$ |
| $\mathbf{I}_N$ | The identity matrix of size $N \times N$ |
| $E[\cdot]$ and $Var[\cdot]$ | The expected value and variance of a random variable, respectively |
| $\mathbf{X}^{\mathcal{H}}$ | The conjugate transpose of $\mathbf{X}$ |
| $\mathcal{Q}(x)$ | The value of the right tail function of the standard normal distribution at $x$ |
| $\mathcal{Q}(x)^{-1}$ | The inverse function of the $\mathcal{Q}(x)$ |
| $R_{sum}(N, \varepsilon)$ | Achievable sum rate of the SISO-FMAC-DMS |
| $R_{sum}^{\star}(N, \varepsilon)$ | The maximum sum-rate of the SISO-FMAC-DMS |
| $R_{sum-miso}(N, \varepsilon)$ | Achievable sum rate of the MISO-FMAC-DMS |
| $R_{sum-miso}^{\star}(N, \varepsilon)$ | The maximum sum-rate of the MISO-FMAC-DMS |
| $W_a$ and $W_b$ | Transmitted messages |
| $P_a$, $P_b$ and $P^{\star}$ | Power constraints |
| $\mathbf{h}_a$, $\mathbf{h}_b$, $\mathbf{g}_a$ and $\mathbf{g}_b$ | Channel gains |
| $P_a$ and $P_b$ | Power constraints |
| $\varepsilon_n$ and $\varepsilon_n'$ | Estimation errors of messages |
| $\Delta$ | Normalized equivocation of the eavesdropper characterizes the secrecy level |
| $\delta$ | Secrecy level |
| $(X_{a,n}, X_{b,n})$ and $Y_n$ | The channel outputs and the channel input, respectively |
| $\rho$ | Correlation coefficient of $X_{a,n}$ and $X_{b,n}$ |

eavesdropper's channel is known by the eavesdropper and the transceiver. In addition, we assume that the eavesdropper also knows the perfect CSI of the legitimate receiver's channel.

The model formulation for the SISO-FMAC-DMS is given in Subsection 3.1, see details below.

## 3.1 Model formulation of the SISO-FMAC-DMS

For the SISO-FMAC-DMS (see Fig 1), the $n$-th ($n = 1, 2, \ldots, N$) channel inputs and output relationship is given by

$$Y_n = h_a X_{a,n} + h_b X_{b,n} + \eta_n \tag{1}$$

here note that $Y_n \in \mathbb{C}^{1 \times 1}$, $X_{a,n} \in \mathbb{C}^{1 \times 1}$, $X_{b,n} \in \mathbb{C}^{1 \times 1}$, and $\eta_n \in \mathbb{C}^{1 \times 1} \sim \mathcal{CN}(0, \sigma^2)$.

**Definition 1**. $X_{a,n}$ and $X_{b,n}$ are the outputs of the channel encoders subject to the average power constraints

$$\frac{1}{N} \sum_{n=1}^{N} E[X_{a,n} \bar{X}_{a,n}] \leq P_a, \qquad \frac{1}{N} \sum_{n=1}^{N} E[X_{b,n} \bar{X}_{b,n}] \leq P_b. \tag{2}$$

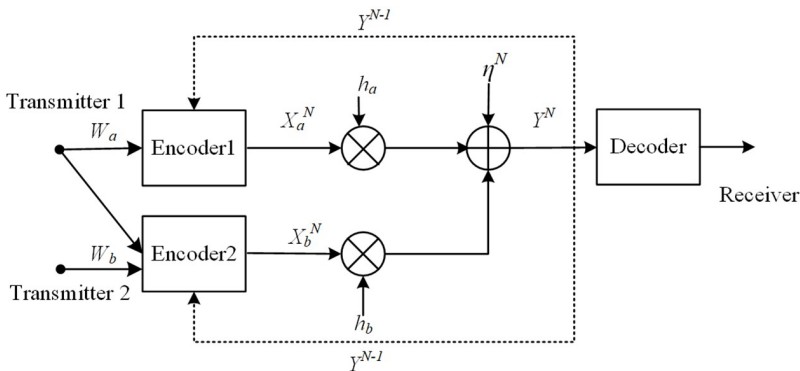

**Fig 1. The model of SISO-FMAC-DMS.**

The messages $W_a$ and $W_b$ are uniformly distributed on finite sets $\{1, 2, ..., |\mathcal{W}_a|\}$ and $\{1, 2, ..., |\mathcal{W}_b|\}$, respectively. The legitimate receiver generates an estimation $\hat{W}_a = \psi_a(Y^N)$ and $\hat{W}_b = \psi_b(Y^N)$, where $\psi_a$ and $\psi_b$ are the legitimate receiver's decoding function.

**Definition 2**. The average decoding error probability equals

$$P_e = \frac{1}{|\mathcal{W}_a||\mathcal{W}_b|} \sum_{w_a \in \mathcal{W}_a, w_b \in \mathcal{W}_b} Pr\{\psi(Y^N) \neq (w_a, w_b)|(w_a, w_b) sent\}. \tag{3}$$

**Definition 3**. The $(N, \varepsilon)$-rate pair $(R_a(N, \varepsilon), R_b(N, \varepsilon))$ is achievable if for given blocklength $N$ and decoding error probability $\varepsilon$, there exists a $(N, |\mathcal{W}_a|, |\mathcal{W}_b|, P_a, P_b)$-code satisfying Definition 1 such that

$$\frac{\log |\mathcal{W}_a|}{N} = R_a(N, \varepsilon), \qquad \frac{\log |\mathcal{W}_b|}{N} = R_b(N, \varepsilon), \qquad P_e \leq \varepsilon. \tag{4}$$

For the SISO-FMAC-DMS, the achievable sum rate is denoted by

$$R_{sum}(N, \varepsilon) = R_a(N, \varepsilon) + R_b(N, \varepsilon), \tag{5}$$

and the maximal sum-rate, denoted as $R_{sum}^\star(N, \varepsilon)$, corresponds to the maximum sum-rate $R_{sum}(N, \varepsilon)$ defined in (5). Then, the model formulation for the MISO-FMAC-DMS is given in Subsection 3.2, see details below.

### 3.2 Model formulation of the MISO-FMAC-DMS

For the MISO-FMAC-DMS (see Fig 2), the transmitter is equipped with $A$ and $B$ antennas, respectively, while the receiver is equipped with a single antenna. The channel fading coefficients are represented by $\mathbf{h}_a \in \mathbb{C}^{1 \times A}$ and $\mathbf{h}_b \in \mathbb{C}^{1 \times B}$. At time $n$ ($n \in \{1, 2, \ldots, N\}$), the channel inputs and output relationship is given by

$$Y_n = \mathbf{h}_a \mathbf{X}_{a.n} + \mathbf{h}_b \mathbf{X}_{b.n} + \eta_n \tag{6}$$

here note that $Y_n \in \mathbb{C}^{1 \times 1}$, $\mathbf{X}_{a,n} \in \mathbb{C}^{A \times 1}$, $\mathbf{X}_{b,n} \in \mathbb{C}^{B \times 1}$, and $\eta_n \in \mathbb{C}^{1 \times 1} \sim \mathcal{CN}(0, \sigma^2)$.

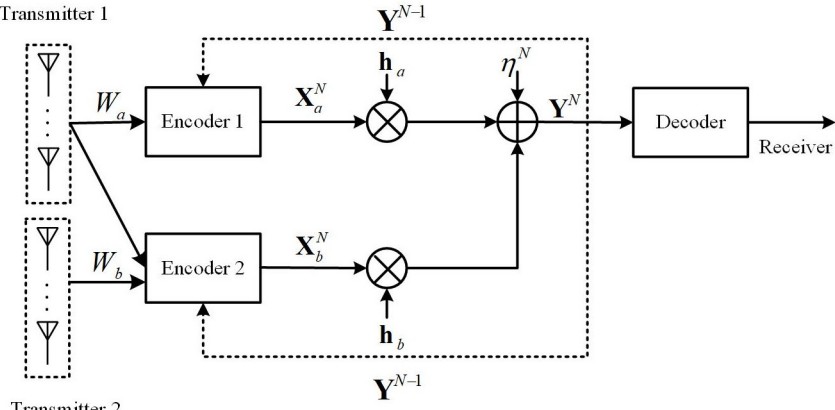

**Fig 2. The model of MISO-FMAC-DMS.**

**Definition 4**. $\mathbf{X}_{a,n}$ and $\mathbf{X}_{b,n}$ are the outputs of the channel encoders subject to the average power constraint

$$\frac{1}{N}\sum_{n=1}^{N}E[||\mathbf{X}_{a,n}||^2] \leq P_a, \qquad \frac{1}{N}\sum_{n=1}^{N}E[||\mathbf{X}_{b,n}||^2] \leq P_b. \tag{7}$$

The messages $W_a$ and $W_b$ are uniformly distributed on finite sets $\{1, 2, ..., |\mathcal{W}_a|\}$ and $\{1, 2, ..., |\mathcal{W}_b|\}$, respectively. The legitimate receiver generates an estimation $\hat{W}_a = \psi_a(Y^N)$ and $\hat{W}_b = \psi_b(Y^N)$, where $\psi_a$ and $\psi_b$ are the legitimate receiver's decoding function.

**Definition 5**. The average decoding error probability is the same that of the Definition 2.

**Definition 6**. The $(N, \varepsilon)$-rate pair $(R_{a-miso}(N, \varepsilon), R_{b-miso}(N, \varepsilon))$ is achievable if for given blocklength $N$ and decoding error probability $\varepsilon$ there exists a $(N, |\mathcal{W}_a|, |\mathcal{W}_b|, P_a, P_b)$-code satisfying Definition 4 such that

$$\frac{\log|\mathcal{W}_a|}{N} = R_{a-miso}(N, \varepsilon), \qquad \frac{\log|\mathcal{W}_b|}{N} = R_{b-miso}(N, \varepsilon), \quad P_e \leq \varepsilon. \tag{8}$$

For the MISO-FMAC-DMS, the achievable sum rate is denoted by

$$R_{sum-miso}(N, \varepsilon) = R_{a-miso}(N, \varepsilon) + R_{b-miso}(N, \varepsilon). \tag{9}$$

and the maximal sum-rate, denoted as $R^{\star}_{sum-miso}(N, \varepsilon)$, corresponds to the maximum sum-rate $R_{sum-miso}(N, \varepsilon)$ defined in (9).

## 4 SK feedback scheme for the FMAC-DMS

The SK feedback coding schemes for the SISO-FMAC-DMS is given in Subsection 4.1, see details below.

### 4.1 Main results of the SISO-FMAC-DMS

**Theorem 1**. For given decoding error probability $\varepsilon$ and codeword length $N$, the maximal sum-rate of the SISO-FMAC-DMS is lower bounded by

$$R^{\star}_{sum}(N, \varepsilon) \geq R_{sum}(N, \varepsilon), \tag{10}$$

where

$$R_{sum}(N,\varepsilon) = \max_{\rho\in[0,1]} \log M_a M_b - \frac{1}{N}\log\left(\left[Q^{-1}\left(\frac{\varepsilon}{8}\right)\right]^4 \frac{(b^2 P^* + r^2)r^2}{9b^4 P^* P_w(1-\rho^2)}\right), \tag{11}$$

here note that

$$P^* = \frac{a^2}{b^2}P_v + \rho^2 P_w + 2\sqrt{\frac{a}{b}P_v P_w}\rho, \qquad P_v = \frac{1}{2}P_a, \qquad P_w = \frac{1}{2}P_b,$$

$$\rho = \frac{E[X_a X_b]}{\sqrt{P_a P_b}}, \qquad r = \sqrt{b^2(1-\rho^2)P_v + \sigma_1^2}, \qquad M_a = 1 + \frac{b^2 P^*}{r^2}, \qquad M_b = 1 + \frac{b^2 P_w(1-\rho^2)}{\sigma_1^2}, \tag{12}$$

and $a$ and $b$ are the magnitudes of $h_a$ and $h_b$, respectively.

**Proof**:

Let

$$X_{a,n} = \frac{\bar{h}_a}{a}X'_{a,n} \Rightarrow X'_{a,n} = \frac{h_a X_{a,n}}{a},$$

$$X_{b,n} = \frac{\bar{h}_b}{b}X'_{b,n} \Rightarrow X'_{b,n} = \frac{h_b X_{b,n}}{b}, \tag{13}$$

where $a$ and $b$ represent the magnitudes of $h_a$ and $h_b$, respectively.

The power constraints of $X'_{a,n}$ and $X'_{b,n}$ are given by

$$E[X'_{a,n}\bar{X}'_{a,n}] = P_a, \qquad E[X'_{b,n}\bar{X}'_{b,n}] = P_b, \tag{14}$$

which indicates that the power constraints of $X'_{a,n}$ and $X'_{b,n}$ are equivalent to those of the $X_{a,n}$ and $X_{b,n}$, respectively. Therefore, the input-output relationship of the channel can be described as follows

$$Y_n = h_a X_{a,n} + h_b X_{b,n} + \eta_n \Rightarrow Y_n = aX'_{a,n} + bX'_{b,n} + \eta_n. \tag{15}$$

By channel splitting, (15) is equivalent to

$$Y_{R,n} + jY_{I,n} = a(X_{aR,n} + jX_{aR,n}) + b(X_{bR,n} + jX_{bR,n}) + \eta_{R,n} + j\eta_{I,n},$$

$$\Rightarrow \quad Y_{R,n} = aX_{aR,n} + bX_{bR,n} + \eta_{R,n}, \quad Y_{I,n} = aX_{aI,n} + bX_{bI,n} + \eta_{I,n}, \tag{16}$$

where $X_{aR,n}$ and $X_{aI,n}$ are the real part and imaginary part of $X'_{a,n}$, respectively, and the other items are analogous. Here let $E(X_{aR,n})^2 = P_{aR} = E(X_{aI,n})^2 = P_{aI} = \frac{1}{2}P_a = P_v$, $E(X_{bR,n})^2 = P_{bR} = E(X_{bI,n})^2 = P_{bI} = \frac{1}{2}P_b = P_w$, $Var[\eta_{R,n}] = Var[\eta_{I,n}] = \frac{\sigma^2}{2} = \sigma_1^2$.

*Brief introduction about the proposed two-step SK coding scheme*: Fig 3 plot the equivalent channel of two-step SK encoding. The two encoders encode the common message $W_a$ and private message $W_b$, respectively. Encoder 1 encodes the message $W_a$ and feedback $Y^{N-1}$ into $X_a^N$ with power $P_a$. Encoder 2 encodes the message $W_a$ and feedback $Y^{N-1}$ into $U^N$ with power $\rho^2 P_b$, and encodes the message $W_b$ and feedback $Y^{N-1}$ into $V^N$ with power $(1-\rho^2)P_b$, where $0 \leq \rho \leq 1$. The receiver decodes $U^N$ and $X_a^N$ first, and then subtracts them from the received message to further decode $V^N$. Therefore, when Encoder 2 encodes $W_b$, the influence of message $W_a$ can be completely eliminated. This indicates that for message $W_b$, the encoding equivalent channel is single-input single-output with AWGN noise. As for the encoding process for message $W_a$, the input to the encoding equivalent channel is $bU^N + aX_a^N$, and the noise of the channel is non-Gaussian white noise $\eta^N + bV^N$.

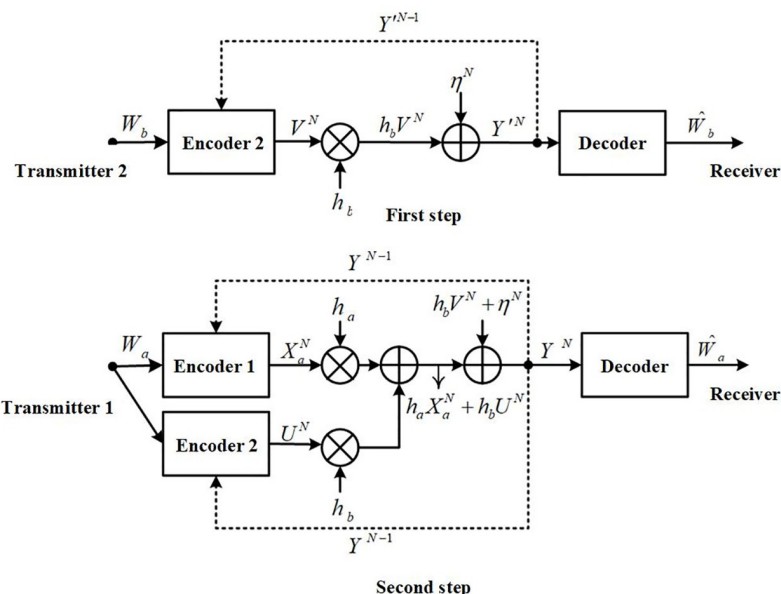

**Fig 3. The steps of the SISO-FMAC-DMS two-step coding scheme.**

Since the expressions of the two sub-channels are similar, analyzing one of them is sufficient. Doubling the achievable sum rate obtained from the analysis will give the achievable rate of the original channel. The following is an introduction to the encoding and decoding process for the real part of the subchannel. The expression for the output of the real part subchannel at time $n(n \in \{1, 2, \ldots, N\})$ is as follows

$$
\begin{aligned}
Y_{R,n} &= aX_{aR,n} + b(U_{R,n} + V_{R,n}) + \eta_{R,n} \\
\Rightarrow \frac{Y_{R,n}}{b} &= \left(\frac{a}{b}X_{aR,n} + U_{R,n}\right) + V_{R,n} + \frac{\eta_{R,n}}{b}.
\end{aligned}
\tag{17}
$$

Define

$$
X_n^* = \frac{a}{b}X_{aR,n} + U_{R,n}, \qquad \eta_{R,n}' = V_{R,n} + \frac{\eta_{R,n}}{b}.
\tag{18}
$$

The expression for the real part sub-channel at time n is $X_n^* \sim \mathcal{N}(0, P^*)$, where $P^*$ is given by

$$
P^* = \frac{a^2}{b^2}P_v + \rho^2 P_w + 2\sqrt{\frac{a}{b}P_v P_w}\rho\rho_n' \leq \frac{a^2}{b^2}P_v + \rho^2 P_w + 2\sqrt{\frac{a}{b}P_v P_w}\rho = P^*,
\tag{19}
$$

where $\rho = \frac{E[X_a X_b]}{\sqrt{P_a P_b}}$, $0 \leq \rho_n' \leq 1$ and $\rho_n'$ is given by

$$
\rho_n' = \frac{E\left[\frac{a}{b}X_{aR,n} U_{R,n}\right]}{\rho\sqrt{\frac{a}{b}P_v P_w}} = \frac{E[X_{aR,n} U_{R,n}]}{\rho\sqrt{P_v P_w}} \cdot \sqrt{\frac{a}{b}},
\tag{20}
$$

and $X_{aR,n} \sim \mathcal{N}(0, P_v)$, $U_{R,n} \sim \mathcal{N}(0, (1 - \rho^2)P_w)$, the relationship between $U_{R,n}$ and $X_{aR,n}$ is

given by

$$U_{R,n} = \rho \sqrt{\frac{P_w}{P_v}} X_{aR,n}. \tag{21}$$

*Message mapping*: Dividing $W_a$ into two parts $(W_{aR}, W_{aI})$, where $W_{aR}$ and $W_{aI}$ take values in $\mathcal{W}_{aR} = \{1, 2, ..., 2^{NR_{aR}(N,\varepsilon)}\}$ and $\mathcal{W}_{aI} = \{1, 2, ..., 2^{NR_{aI}(N,\varepsilon)}\}$, respectively. Then divide the interval [-0.5,0.5] into $2^{NR_{aR}}(N, \varepsilon)$ equally spaced sub-intervals, and the center of each sub-interval is mapped to a message value in $\mathcal{W}_{aR}$. Let $\theta_1$ be the center of the sub-interval with respect to the message $W_{aR}$. Since $W_{aR}$ is equi-probably distributed over the set $\mathcal{W}_{aR}$, $\theta_1$ is approximately uniformly distributed over the interval $[-0.5, 0.5]$ and its variance approximately equals $\frac{1}{12}$, i.e. $E[\theta_1]^2 = \frac{1}{12}$. $W_b$, $W_{bR}$ and $\theta_2$ are similar to $W_a$, $W_{aR}$ and $\theta_1$, respectively, hence we omit them here.

At time 1, the encoder 1 transmits

$$X_{aR,1} = 0, \tag{22}$$

the encoder 2 transmits

$$V_{R,1} = \sqrt{12(1 - \rho^2)P_w}\theta_2, \tag{23}$$

and

$$U_{R,1} = \rho \sqrt{\frac{P_w}{P_v}} X_{aR,1} = 0. \tag{24}$$

The receiver receives $\frac{Y'_{R,1}}{b}$, and obtains an estimation of $\theta_2$ by computing

$$\hat{\theta}_{2,1} = \frac{Y_{R,1}'}{b\sqrt{12(1 - \rho^2)P_w}} = \theta_2 + \frac{\eta_{R,1}}{b\sqrt{12(1 - \rho^2)P_w}} = \theta_2 + \varepsilon_1', \tag{25}$$

where $\varepsilon_1' = \frac{\eta_{R,1}}{b\sqrt{12(1-\rho^2)P_w}}$, and $\alpha_1' \triangleq Var(\varepsilon_1) = \frac{\sigma_1^2}{12b^2(1-\rho^2)P_w}$.

At time 2, the encoder 1 transmits $X_{aR,2}$, encoder 2 transmits

$$U_{R,2} = \rho \sqrt{\frac{P_w}{P_v}} X_{aR,2}, \qquad V_{R,2} = \sqrt{\frac{(1 - \rho^2)P_w}{\alpha_1'}}\varepsilon_1', \tag{26}$$

$X_2^*$ is given by

$$X_2^* = \frac{a}{b} X_{aR,2} + U_{R,2} = \sqrt{12P^*}\theta_1. \tag{27}$$

the receiver obtains $\frac{Y_{R,2}}{b}$, and gets an estimation of $\theta_1$ by computing

$$\frac{Y_{R,2}}{b} = X_2^* + V_{R,2} + \frac{\eta_{R,2}}{b} = \sqrt{12P^*}\theta_1 + \left(V_{R,2} + \frac{\eta_{R,2}}{b}\right),$$

$$\Rightarrow \hat{\theta}_{1,1} = \frac{Y_{R,2}}{b\sqrt{12P^*}} = \theta_1 + \frac{V_{R,2} + \eta_{R,2}}{b\sqrt{12P^*}} = \theta_1 + \varepsilon_2, \tag{28}$$

where $\varepsilon_2 = \frac{V_{R,2}+\eta_{R,2}}{b\sqrt{12P^*}}$ and $\alpha_2 \triangleq Var(\varepsilon_2) = \frac{b^2(1-\rho^2)P_w+\sigma_1^2}{12b^2P^*}X_{aR,2}$.

At time $n$ ($3 \le n \le N$), the transmitter 1 transmits $X_{aR,n}$ and $U_{R,n} = \rho \sqrt{\frac{P_w}{P_v}} X_{aR,n}$, $X_n^*$ is given by

$$X_n^* = U_{R,n} + \frac{X_{aR,n}}{b} = \sqrt{\frac{P^*}{\alpha_{n-1}}} \varepsilon_{n-1}, \tag{29}$$

and transmitter 2 transmits

$$V_{R,n} = \sqrt{\frac{(1-\rho^2)P_w}{\alpha_{n-1}'}} \varepsilon_{n-1}'. \tag{30}$$

The receiver obtains $\frac{Y_{R,n}}{b}$, and gets the estimation of $\theta_{1,n}$ and $\theta_{2,n}$ by computing

$$
\begin{aligned}
\hat{\theta}_{1,n} &= \hat{\theta}_{1,n-1} - \frac{E\left[\frac{Y_{R,n}}{b}\varepsilon_{n-1}\right]}{E\left[\left(\frac{Y_{R,n}}{b}\right)^2\right]} \frac{Y_{R,n}}{b}, \\
\hat{\theta}_{2,n} &= \hat{\theta}_{2,n-1} - \frac{E\left[\frac{Y'_{R,n}}{b}\varepsilon'_{n-1}\right]}{E\left[\left(\frac{Y'_{R,n}}{b}\right)^2\right]} \frac{Y'_{R,n}}{b},
\end{aligned}
\tag{31}
$$

where

$$
\begin{aligned}
\varepsilon_{n-1} &= \varepsilon_{n-2} - \frac{E\left[\frac{Y_{R,n-1}}{b}\varepsilon_{n-2}\right]}{E\left[\left(\frac{Y_{R,n-1}}{b}\right)^2\right]} \frac{Y_{R,n-1}}{b} = \hat{\theta}_{1,n-1} - \theta_1, \\
\varepsilon_{n-1}' &= \varepsilon_{n-2}' - \frac{E\left[\frac{Y_{R,n-1}'}{b}\varepsilon_{n-2}'\right]}{E\left[\left(\frac{Y_{R,n-1}'}{b}\right)^2\right]} \frac{Y_{R,n-1}'}{b} = \hat{\theta}_{2,n-1} - \theta_2.
\end{aligned}
\tag{32}
$$

**Lemma 1:** For given decoding error probability $\varepsilon$ and codeword length $N$, if the transmission rates $R_a(N, \varepsilon)$ and $R_b(N, \varepsilon)$ satisfy

$$
\begin{aligned}
R_a(N, \varepsilon) &= \log\left(1 + \frac{b^2 P^*}{r^2}\right) - \frac{1}{N}\log\left(\frac{b^2 P^* + r^2}{3b^2 P^*} \cdot \left[Q^{-1}\left(\frac{\varepsilon}{8}\right)\right]^2\right), \\
R_b(N, \varepsilon) &= \log\left(1 + \frac{b^2(1-\rho^2)P_w}{\sigma_1^2}\right) - \frac{1}{N}\log\left(\frac{b^2(1-\rho^2)P_w + \sigma_1^2}{3b^2(1-\rho^2)P_w} \cdot \left[Q^{-1}\left(\frac{\varepsilon}{8}\right)\right]^2\right),
\end{aligned}
\tag{33}
$$

the decoding error probability satisfy $P_e \le \varepsilon$.

**Proof:** See S1 Appendix.

Finally, from (33), the achievable sum rate $R_{sum}(N, \varepsilon)$ given in (11) is obtained. The details about SK feedback coding scheme for the MISO-FMAC-DMS are given in Subsection 4.2.

## 4.2 Main results of MISO-FMAC-DMS

**Theorem 2**. For given decoding error probability $\varepsilon$ and codeword length $N$, the maximal sum-rate of the MISO-FMAC-DMS is lower bounded by

$$R^*_{sum-miso}(N, \varepsilon) \geq R_{sum-miso}(N, \varepsilon), \tag{34}$$

where

$$R_{sum-miso}(N, \varepsilon) = \max_{\rho \in [0,1]} \log M_1 M_2 - \frac{1}{N} \log \left( \left[ Q^{-1} \left( \frac{\varepsilon}{8} \right) \right]^4 \cdot \frac{(\tilde{P} + t^2)^2}{9 ||\mathbf{h}_b||^2 \tilde{P} P_2 (1 - \rho^2)} \right), \tag{35}$$

and

$$M_1 = 1 + \frac{\tilde{P}}{t^2}, \qquad M_2 = 1 + \frac{||h_2||^2 P_2 (1 - \rho^2)}{\sigma_1^2},$$

$$t = \sqrt{||\mathbf{h}_b||^2 (1 - \rho^2) P_2 + \sigma_1^2}, \tag{36}$$

$$\tilde{P} = ||\mathbf{h}_a||^2 P_1 + ||\mathbf{h}_b||^2 \rho^2 P_2 + 2 \sqrt{||\mathbf{h}_a|| \cdot ||\mathbf{h}_b|| P_1 P_2} \rho,$$

and $\rho$ is given in Theorem 1.

**Proof:**

Transmit beamforming strategy for the feedforward channel: In (6), letting

$$\mathbf{X}_{a,n} = \frac{\mathbf{h}_a^{\mathcal{H}}}{||\mathbf{h}_a||} X_{a,n} \Rightarrow X_{a,n} = \frac{\mathbf{h}_a \mathbf{X}_{a,n}}{||\mathbf{h}_a||},$$

$$\mathbf{X}_{b,n} = \frac{\mathbf{h}_b^{\mathcal{H}}}{||\mathbf{h}_b||} X_{b,n} \Rightarrow X_{b,n} = \frac{\mathbf{h}_b \mathbf{X}_{b,n}}{||\mathbf{h}_b||} \tag{37}$$

here note that $\mathbf{X}_{a,n} \in \mathbb{C}^{A \times 1}$, $\mathbf{X}_{b,n} \in \mathbb{C}^{B \times 1}$, $\mathbf{h}_a \in \mathbb{C}^{1 \times A}$, $\mathbf{h}_b \in \mathbb{C}^{1 \times B}$.

Therefore, the input-output relationship of the channel can be described as

$$Y_n = \mathbf{h}_a \mathbf{X}_{a,n} + \mathbf{h}_b \mathbf{X}_{b,n} + \eta_n$$

$$\Rightarrow \quad Y_n = ||\mathbf{h}_a|| X_{a,n} + ||\mathbf{h}_b|| X_{b,n} + \eta_n, \tag{38}$$

By channel splitting, we have

$$Y_{R,n} + j Y_{I,n} = ||\mathbf{h}_a|| (X_{aR,n} + j X_{aI,n}) + ||\mathbf{h}_b|| (X_{bR,n} + j X_{bI,n}) + \eta_{R,n} + j \eta_{I,n}$$

$$\Rightarrow Y_{R,n} = ||\mathbf{h}_a|| X_{aR,n} + ||\mathbf{h}_b|| X_{bR,n} + \eta_{R,n}, \tag{39}$$

$$Y_{I,n} = ||\mathbf{h}_a|| X_{aI,n} + ||\mathbf{h}_b|| X_{bI,n} + \eta_{I,n},$$

and the power constraints are given below, $E(X_{aR,n})^2 = P_{aR} = E(X_{aI,n})^2 = P_{aI} = \frac{1}{2} P_a = P_1$, $E(X_{bR,n})^2 = P_{bR} = E(X_{bI,n})^2 = P_{bI} = \frac{1}{2} P_b = P_2$, $Var[\eta_{R,n}] = Var[\eta_{I,n}] = \frac{\sigma^2}{2} = \sigma_1^2$.

Define

$$\tilde{X}_n = \frac{a}{b} X_{aR,n} + U_{R,n}, \qquad \eta_{R,n}{}' = V_{R,n} + \frac{\eta_{R,n}}{b}. \tag{40}$$

The expression for the real part sub-channel at time n is $\tilde{X}_n \sim \mathcal{N}(0, \tilde{P})$, where $\tilde{P}$ is given by

$$\tilde{P} = ||\mathbf{h}_a||^2 P_1 + ||\mathbf{h}_b||^2 \rho^2 P_2 + 2\sqrt{||\mathbf{h}_a|| \cdot ||\mathbf{h}_b|| P_1 P_2} \rho \tilde{\rho}_n$$

$$\leq ||\mathbf{h}_a||^2 P + ||\mathbf{h}_b||^2 \rho^2 P_2 + 2\sqrt{||\mathbf{h}_a|| \cdot ||\mathbf{h}_b|| P_1 P_2} \rho = \tilde{P}, \tag{41}$$

where $\rho = \frac{E[X_a X_b]}{\sqrt{P_a P_b}}$, $0 \leq \tilde{\rho}_n \leq 1$, and $\tilde{\rho}_n$ is given by

$$\tilde{\rho}_n = \frac{E[X_{aR,n} U_{R,n}]}{\rho\sqrt{P_1 P_2}} \cdot \sqrt{||\mathbf{h}_a|| \cdot ||\mathbf{h}_b||}, \tag{42}$$

and $X_{aR,n} \sim \mathcal{N}(0, P_1)$, $U_{R,n} \sim \mathcal{N}(0, (1 - \rho^2)P_2)$, the relationship between $U_{R,n}$ and $X_{aR,n}$ is given by

$$U_{R,n} = \rho\sqrt{\frac{P_2}{P_1}} X_{aR,n}. \tag{43}$$

The encoding-decoding procedure of our proposed SK-type scheme for the MISO-F-MAC-DMS of Fig 4 is described.

Along the lines of the encoding-decoding procedure in Section 4.1, we conclude that the transmission rates $R_{aR-miso}(N, \varepsilon)$ and $R_{bR-miso}(N, \varepsilon)$ can be expressed as

$$R_{aR-miso}(N, \varepsilon) = \frac{1}{2} \log\left(1 + \frac{\tilde{P}}{t^2}\right) - \frac{1}{2N} \log\left(\frac{\tilde{P} + t^2}{3\tilde{P}} \cdot \left[Q^{-1}\left(\frac{\varepsilon}{8}\right)\right]^2\right),$$

$$R_{bR-miso}(N, \varepsilon) = \frac{1}{2} \log\left(1 + \frac{||\mathbf{h}_b||^2 (1 - \rho^2) P_2}{\sigma_1^2}\right) \tag{44}$$

$$- \frac{1}{2N} \log\left(\frac{||\mathbf{h}_b||^2 (1 - \rho^2) P_2 + \sigma_1^2}{3b^2 (1 - \rho^2) P_2} \cdot \left[Q^{-1}\left(\frac{\varepsilon}{8}\right)\right]^2\right).$$

Therefore, the sum rate $R_{sum-miso}(N, \varepsilon)$ given in (35) is obtained, and we complete the proof.

## 5 A secure FBL SK-type coding scheme for the MISO-FMAC-DMS with an external eavesdropper

The model formulation for MISO-FMAC-DMS with an external eavesdropper is given in Subsection 5.1, see details below.

### 5.1 Model formulation for MISO-FMAC-DMS with an external eavesdropper

The information-theoretic schematic diagram for the MISO-FMAC-DMS with an external eavesdropper is almost the same as that in Subsection 3.2 (see Fig 5), except that the existence of an eavesdropper equipped with $C$ ($C \geq 1$) antennas, the channel gains of eavesdropping channels for Transmitter 1 and Transmitter 2 are denoted by $\mathbf{g}_a \in \mathbb{C}^{A \times C}$, $\mathbf{g}_b \in \mathbb{C}^{B \times C}$, respectively, and they are mutually independent of each other and stay constants during the transmission.

At time $n$-th ($n \in \{1, 2, \ldots, N\}$), the signal received by the eavesdropper is given by

$$\mathbf{Z}_n = \mathbf{g}_a \mathbf{X}_{a,n} + \mathbf{g}_b \mathbf{X}_{b,n} + \zeta_n, \tag{45}$$

where $\zeta_n \sim \mathcal{CN}(0, \sigma_\zeta^2)$.

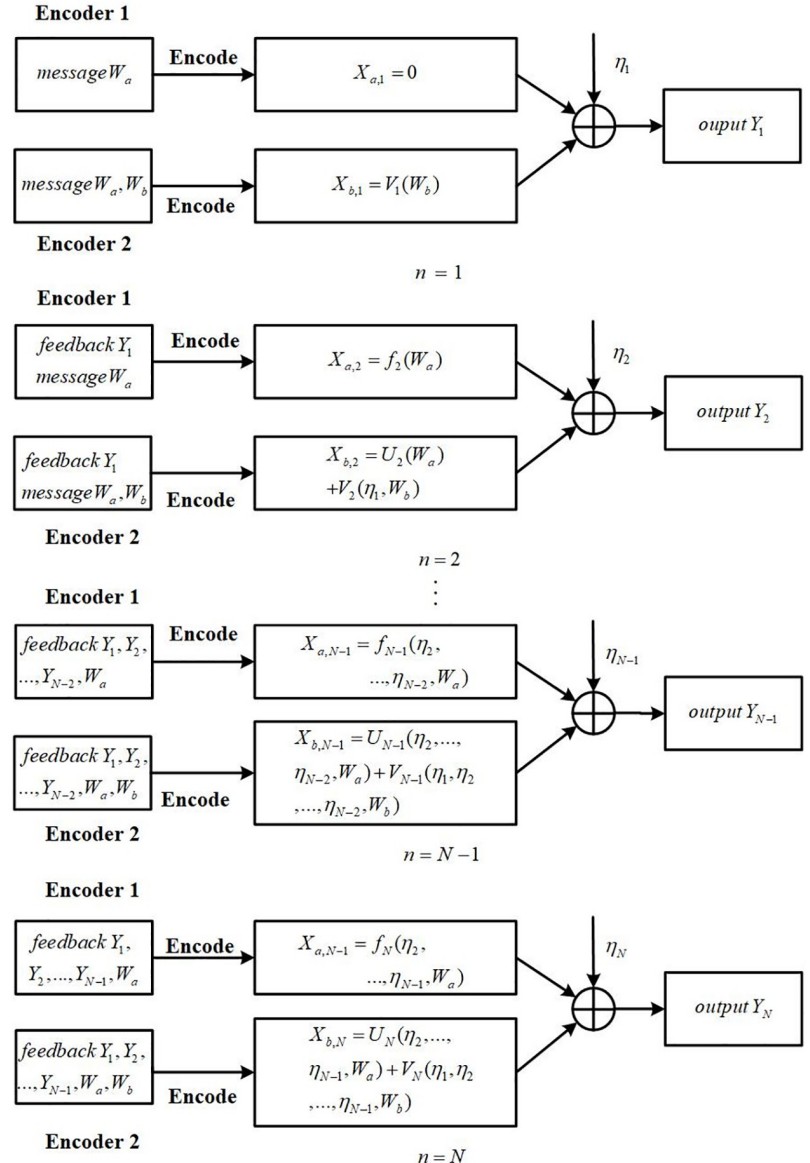

**Fig 4. Procedure of our proposed SK-type scheme for the MISO-FMAC-DMS.**

**Definiton 7**. By using the secrecy criteria defined in [14, 15], in this paper, a normalized equivocation of the eavesdropper characterizes the secrecy level and it is given by

$$\Delta = \frac{H(W_a, W_b | \mathbf{Z}^N, \mathbf{h}, \mathbf{g})}{H(W_a, W_b)}, \quad 0 \leq \Delta \leq 1, \tag{46}$$

where $\mathbf{h} = (\mathbf{h}_a, \mathbf{h}_b)$, $\mathbf{g} = (\mathbf{g}_a, \mathbf{g}_b)$.

The rate pair $(R_{a-miso}(N, \varepsilon, \delta), R_{b-miso}(N, \varepsilon, \delta))$ is said to be achievable with $\delta$-secrecy, if for given blocklength $N$, decoding error probability $\varepsilon$ there exists a $(N, |W_a|, |W_b|, P_a, P_b)$-code

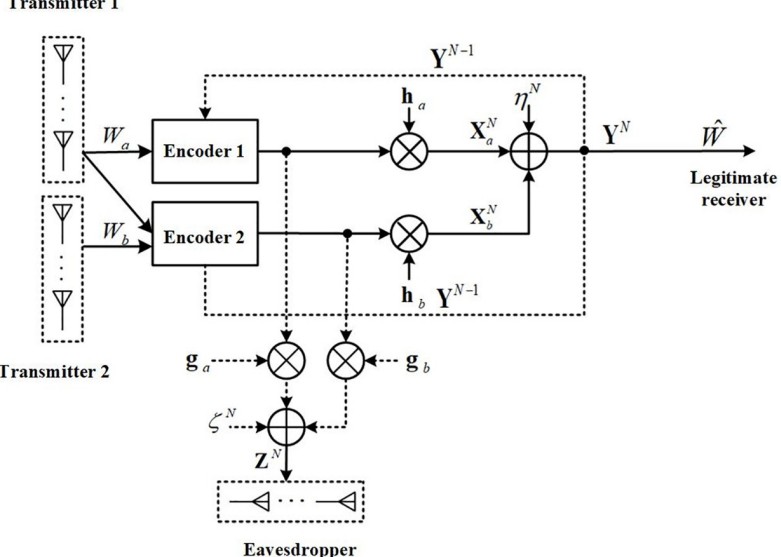

**Fig 5. The MISO-FMAC-DMS with an external eavesdropper.**

described in Definition 4 such that

$$\frac{\log |W_a|}{N} = R_{a-miso}(N, \varepsilon, \delta), \qquad \frac{\log |W_b|}{N} = R_{b-miso}(N, \varepsilon, \delta), P_e \le \varepsilon, \quad \Delta \ge \delta. \qquad (47)$$

Here note that $0 \le \delta \le 1$ is the required secrecy level. $\delta = 1$ corresponds to perfect secrecy. For given $N$, $\varepsilon$ and $\delta$, a $(N, |W_a|, |W_b|, P_a, P_b)$-code defined in Definition 4 is said to be secure if (47) is satisfied, and the achievable secrecy sum rate is denoted by

$$R_{sum-miso}(N, \varepsilon, \delta) = R_{a-miso}(N, \varepsilon, \delta) + R_{b-miso}(N, \varepsilon, \delta). \qquad (48)$$

Then, we show that our proposed scheme for the MISO-FMAC-DMS is also a secure FBL scheme when the coding blocklength is lager than a certain threshold in Subsection 5.2.

## 5.2 A secure FBL SK-type coding scheme for the MISO-FMAC-DMS with an external eavesdropper

**Theorem 3**. For given decoding error probability $\varepsilon$ and secrecy level $\delta$, if the blocklength $N$ satisfies

$$N \ge \frac{\frac{M_{12}}{1-\delta} + \log\left(\left[Q^{-1}\left(\frac{\varepsilon}{8}\right)\right]^4 \cdot \frac{(\tilde{P}+t^2) \cdot t^2}{9\|\mathbf{h}_b\|^2 \tilde{P} P_2 (1-\rho^2)}\right)}{\log M_1 M_2}, \qquad (49)$$

an achievable secrecy sum rate $R_{sum-miso}(N, \varepsilon, \delta)$ of the MISO-FMAC-DMS with an external eavesdropper is given by

$$R_{sum-miso}(N, \varepsilon, \delta) = R_{sum-miso}(N, \varepsilon), \qquad (50)$$

where

$$
\begin{aligned}
M_{12} = \quad & \log \det \left( I_C + \frac{P_1 \mathbf{g}_a \mathbf{h}_a{}^{\mathcal{H}} \mathbf{h}_a \mathbf{g}_a{}^{\mathcal{H}}}{||\mathbf{h}_a||^2 \sigma_\zeta^2} + \frac{\rho^2 P_2 \mathbf{g}_b \mathbf{h}_b{}^{\mathcal{H}} \mathbf{h}_b \mathbf{g}_b{}^{\mathcal{H}}}{||\mathbf{h}_b||^2 \sigma_\zeta^2} \right) \\
& + \log \det \left( I_C + \frac{(1 - \rho^2) P_2 \mathbf{g}_b \mathbf{h}_b{}^{\mathcal{H}} \mathbf{h}_b \mathbf{g}_b{}^{\mathcal{H}}}{||\mathbf{h}_b||^2 \sigma_\zeta^2} \right).
\end{aligned}
\tag{51}
$$

**Proof**:

At time $n$-th ($n \in \{1, 2, \ldots, N\}$), the signal received by the eavesdropper can be rewritten as

$$
\mathbf{Z}_n = \mathbf{g}_a \frac{\mathbf{h}_a{}^{\mathcal{H}}}{||\mathbf{h}_a||} X_{a,n} + \mathbf{g}_b \frac{\mathbf{h}_b{}^{\mathcal{H}}}{||\mathbf{h}_b||} (U_n + V_n) + \zeta_n.
\tag{52}
$$

To calculate the equivocation $\Delta = \frac{H(W_a, W_b | \mathbf{Z}^N, \mathbf{h}, \mathbf{g})}{H(W_a, W_b)}$, we bound $H(W_a, W_b | \mathbf{Z}^N, \mathbf{h}, \mathbf{g})$ first, which is given by

$$
\begin{aligned}
H(W_a, W_b | \mathbf{Z}^N, \mathbf{h}, \mathbf{g}) &= H(W_a | \mathbf{Z}^N, \mathbf{h}, \mathbf{g}) + H(W_b | W_a, \mathbf{Z}^N, \mathbf{h}, \mathbf{g}) \\
&\geq H(W_a | W_b, \mathbf{Z}^N, \mathbf{h}, \mathbf{g}) + H(W_a | W_b, \mathbf{Z}^N, \mathbf{h}, \mathbf{g}).
\end{aligned}
\tag{53}
$$

Next, the term $H(W_a | W_b, \mathbf{Z}^N, \mathbf{h}, \mathbf{g})$ can be further bounded by

$$
\begin{aligned}
& H(W_a | W_b, \mathbf{Z}^N, \mathbf{h}, \mathbf{g}) \\
& \overset{(a)}{\geq} H(W_a | W_b, \mathbf{Z}^N, \mathbf{h}, \mathbf{g}, \eta_1, \eta_2, \ldots, \eta_{N-1}, \zeta_1, \zeta_3, \zeta_4, \ldots, \zeta_N) \\
& \overset{(b)}{\geq} H\Big(W_a \Big| W_b, \mathbf{g}_a \frac{\mathbf{h}_a{}^{\mathcal{H}}}{||\mathbf{h}_a||} X_{a,2} + \mathbf{g}_b \frac{\mathbf{h}_b{}^{\mathcal{H}}}{||\mathbf{h}_b||} (U_2 + V_2) + \zeta_2, \\
& \quad \mathbf{h}, \mathbf{g}, \eta_1, \eta_2, \ldots, \eta_{N-1}, \zeta_1, \zeta_3, \zeta_4, \ldots, \zeta_N \Big) \\
& \overset{(c)}{=} H\left( W_a \Big| \mathbf{g}_a \frac{\mathbf{h}_a{}^{\mathcal{H}}}{||\mathbf{h}_a||} X_{a,2} + \mathbf{g}_b \frac{\mathbf{h}_b{}^{\mathcal{H}}}{||\mathbf{h}_b||} U_2 + \zeta_2 \right) \\
& \overset{(d)}{=} H(W_a) + h(\zeta_2) - h\left( \mathbf{g}_a \frac{\mathbf{h}_a{}^{\mathcal{H}}}{||\mathbf{h}_a||} X_{a,2} + \mathbf{g}_b \frac{\mathbf{h}_b{}^{\mathcal{H}}}{||\mathbf{h}_b||} U_2 + \zeta_2 \right) \\
& \overset{(e)}{=} H(W_a) - \log \det \left( I_C + \frac{P_1 \mathbf{g}_a \mathbf{h}_a{}^{\mathcal{H}} \mathbf{h}_a \mathbf{g}_a{}^{\mathcal{H}}}{||\mathbf{h}_a||^2 \sigma_\zeta^2} + \frac{\rho^2 P_2 \mathbf{g}_b \mathbf{h}_b{}^{\mathcal{H}} \mathbf{h}_b \mathbf{g}_b{}^{\mathcal{H}}}{||\mathbf{h}_b||^2 \sigma_\zeta^2} \right),
\end{aligned}
\tag{54}
$$

where ($a$) follows from conditioning reduce entropy, ($b$) is due to the fact that $\mathbf{Z}^N$ can be

concretely expressed as

$$
\begin{aligned}
\mathbf{Z}_1 &= \mathbf{g}_b \frac{\mathbf{h}_b^{\mathcal{H}}}{||\mathbf{h}_b||} V_n + \zeta_n, \\
\mathbf{Z}_2 &= \mathbf{g}_a \frac{\mathbf{h}_a^{\mathcal{H}}}{||\mathbf{h}_a||} X_{a,2} + \mathbf{g}_b \frac{\mathbf{h}_b^{\mathcal{H}}}{||\mathbf{h}_b||} (U_2 + V_2) + \zeta_2, \\
\mathbf{Z}_3 &= \mathbf{g}_a \frac{\mathbf{h}_a^{\mathcal{H}}}{||\mathbf{h}_a||} X_{a,3} + \mathbf{g}_b \frac{\mathbf{h}_b^{\mathcal{H}}}{||\mathbf{h}_b||} (U_3 + V_3) + \zeta_3, \\
&\cdots \\
\mathbf{Z}_{N-1} &= \mathbf{g}_a \frac{\mathbf{h}_a^{\mathcal{H}}}{||\mathbf{h}_a||} X_{a,N-1} + \mathbf{g}_b \frac{\mathbf{h}_b^{\mathcal{H}}}{||\mathbf{h}_b||} (U_{N-1} + V_{N-1}) + \zeta_2, \\
\mathbf{Z}_N &= \mathbf{g}_a \frac{\mathbf{h}_a^{\mathcal{H}}}{||\mathbf{h}_a||} X_{a,N} + \mathbf{g}_b \frac{\mathbf{h}_b^{\mathcal{H}}}{||\mathbf{h}_b||} (U_N + V_N) + \zeta_N,
\end{aligned}
\tag{55}
$$

when $n = 2, 3, \ldots, N$, $X_{a,N}$ is a function of $(\eta_2, \eta_3, \ldots, \eta_{N-1})$, and $V_N$ is a function of $(\eta_1, \eta_2, \ldots, \eta_{N-1})$, $(c)$ follows from $\left( W_b, \mathbf{g}_b \frac{\mathbf{h}_b^{\mathcal{H}}}{||\mathbf{h}_b||} V_2, \mathbf{h}, \mathbf{g}, \eta_1, \eta_2, \ldots, \eta_{N-1}, \zeta_1, \zeta_3, \zeta_{4,}, \ldots, \zeta_N \right)$ are independent of $(W_a, X_{a,2}, \zeta_2)$, $(d)$ follows from $X_{a,2}$ and $U_2$ is a function of $W_a$, and $W_a$ is independent of $\zeta_1$), $(e)$ follows from

$$
\begin{aligned}
&h\left( \mathbf{g}_a \frac{\mathbf{h}_a^{\mathcal{H}}}{||\mathbf{h}_a||} X_{a,2} + \mathbf{g}_b \frac{\mathbf{h}_b^{\mathcal{H}}}{||\mathbf{h}_b||} U_2 + \zeta_2 \right) - h(\zeta_2) \\
&= \log \det\left( \pi e \left( E\left[ \mathbf{g}_a \frac{\mathbf{h}_a^{\mathcal{H}}}{||\mathbf{h}_a||} X_{a,2} \right] \left[ \mathbf{g}_a \frac{\mathbf{h}_a^{\mathcal{H}}}{||\mathbf{h}_a||} X_{a,2} \right]^{\mathcal{H}} \right) \right. \\
&\quad \left. + E\left( \left[ \mathbf{g}_b \frac{\mathbf{h}_b^{\mathcal{H}}}{||\mathbf{h}_b||} U_2 \right] \left[ \mathbf{g}_b \frac{\mathbf{h}_b^{\mathcal{H}}}{||\mathbf{h}_b||} U_2 \right]^{\mathcal{H}} + \sigma_\zeta^2 I_C \right) \right) - \log \det(\pi e \sigma_\zeta^2 I_C) \\
&= \log \det\left( I_C + \frac{P_1 \mathbf{g}_a \mathbf{h}_a^{\mathcal{H}} \mathbf{h}_a \mathbf{g}_a^{\mathcal{H}}}{||\mathbf{h}_a||^2 \sigma_\zeta^2} + \frac{\rho^2 P_2 \mathbf{g}_b \mathbf{h}_b^{\mathcal{H}} \mathbf{h}_b \mathbf{g}_b^{\mathcal{H}}}{||\mathbf{h}_b||^2 \sigma_\zeta^2} \right).
\end{aligned}
\tag{56}
$$

Analogously, the term $H(W_b \mid W_a, \mathbf{Z}^N, \mathbf{h}, \mathbf{g})$ can be bounded by

$$
\begin{aligned}
H(W_b \mid W_a, \mathbf{Z}^N, \mathbf{h}, \mathbf{g}) \geq{} &H(W_b) - \\
&\log \det\left( I_C + \frac{(1 - \rho^2) P_2 \mathbf{g}_b \mathbf{h}_b^{\mathcal{H}} \mathbf{h}_b \mathbf{g}_b^{\mathcal{H}}}{||\mathbf{h}_b||^2 \sigma_\zeta^2} \right).
\end{aligned}
\tag{57}
$$

Substituting (54) and (57) into $\Delta$, we conclude that

$$
\begin{aligned}
&\frac{H(W_a, W_b \mid \mathbf{Z}^N, \mathbf{h}, \mathbf{g})}{H(W_a, W_b)} \\
&\geq \frac{H(W_a \mid W_b, \mathbf{Z}^N, \mathbf{h}, \mathbf{g}) + H(W_a \mid W_b, \mathbf{Z}^N, \mathbf{h}, \mathbf{g})}{H(W_a, W_b)} \\
&\stackrel{(f)}{\geq} 1 - \frac{M_{12}}{H(W_a) + H(W_b)} \stackrel{(g)}{=} 1 - \frac{M_{12}}{N R_{sum-miso}(N, \varepsilon, \delta)} \\
&\stackrel{(h)}{\geq} 1 - \frac{M_{12}}{N \log M_1 M_2 - \log \left[ Q^{-1}\left( \frac{\varepsilon}{8} \right) \right]^4 \cdot \frac{(\tilde{P} + t^2) \cdot t^2}{9 ||\mathbf{h}_b||^2 \tilde{P} P_2 (1 - \rho^2)}}.
\end{aligned}
\tag{58}
$$

where (*f*) follows from $H(W_a, W_b) = H(W_a) + H(W_b)$ and

$$M_{12} = \log\det\left(I_C + \frac{P_1 \mathbf{g}_a \mathbf{h}_a{}^{\mathcal{H}} \mathbf{h}_a \mathbf{g}_a{}^{\mathcal{H}}}{||\mathbf{h}_a||^2 \sigma_\zeta^2} + \frac{\rho^2 P_2 \mathbf{g}_b \mathbf{h}_b{}^{\mathcal{H}} \mathbf{h}_b \mathbf{g}_b{}^{\mathcal{H}}}{||\mathbf{h}_b||^2 \sigma_\zeta^2}\right) + \log\det\left(I_C + \frac{(1-\rho^2) P_2 \mathbf{g}_b \mathbf{h}_b{}^{\mathcal{H}} \mathbf{h}_b \mathbf{g}_b{}^{\mathcal{H}}}{||\mathbf{h}_b||^2 \sigma_\zeta^2}\right),$$ (*g*) is due to

$H(W_a) + H(W_b) = N(R_{a-miso}(N, \varepsilon, \delta) + R_{b-miso}(N, \varepsilon, \delta)) = NR_{sum-miso}(N, \varepsilon, \delta)$, (*h*) is follows from $R_{sum-miso}(N, \varepsilon, \delta) = R_{sum-miso}(N, \varepsilon)$. From (58), we conclude that $\Delta \geq \delta$ is guaranteed if

$$1 - \frac{M_{12}}{N \log M_1 M_2 - \log\left(\left(Q^{-1}\left(\frac{\varepsilon}{8}\right)\right)^4 \cdot \frac{(\tilde{P} + t^2) \cdot t^2}{9||\mathbf{h}_b||^2 \tilde{P} P_2 (1 - \rho^2)}\right)} \geq \delta,$$ (59)

which can be rewritten as

$$N \geq \frac{\dfrac{M_{12}}{1-\delta} + \log\left(\left(Q^{-1}\left(\dfrac{\varepsilon}{8}\right)\right)^4 \cdot \dfrac{(\tilde{P} + t^2) \cdot t^2}{9||\mathbf{h}_b||^2 \tilde{P} P_2 (1 - \rho^2)}\right)}{\log M_1 M_2},$$ (60)

and the proof is completed.

## 6 Simulation results

Here note that all simulation results are based on an average of 1000 independent channel realizations. Also, Table 3 provides a summary of the parameters used in the simulation.

Figs 6 and 7 show the results of the comparison between the SNR, blocklength and decoding error probability of the LDPC code [15], Turbo code [16], LDGM code [17] and the proposed linear feedback coding scheme for the FMAC-DMS channel. From Figs 6 and 7, we see that the required coding blocklength and SNR for achieving the desired decoding error probability are significantly lower than the LDPC, Turbo, and LDGM codes. This significant reduction is attributed to the doubly exponential decay of the decoding error probability in our scheme as the coding blocklength increases, and error probabilities of other coding schemes only single-exponentially decay to zero as the blocklength length increases, which indicates that our proposed linear feedback coding scheme is a high-reliability and low-latency coding scheme.

Fig 8 plots the relationship between the sum rate, number of transmitting antenna and power constraint of the feedback channels for the MISO-FMAC-DMS, and let blocklength $N = 30$. From Fig 8, we conclude that the sum rate is increasing while the number of transmitting antenna and power constraint of the feedback channel are increasing.

Fig 9 plots the relationship between the FBL sum rate, decoding error probability and blocklength for the MISO-FMAC-DMS. From Fig 9, we conclude that the sum rate is increasing while the blocklength is increasing, and the blocklength reaches a threshold, reducing the

**Table 3. Parameters in simulation.**

| Notations | Values |
|---|---|
| $\rho$ | 0.5 |
| $\mathbf{h}_a$, $\mathbf{h}_b$, $\mathbf{g}_a$ and $\mathbf{g}_b$ | i.i.d. as $\mathcal{CN}(0, 1)$ |
| $A$, $B$ and $C$ | 6 |
| $P_A$ and $P_B$ | 10 |
| $\sigma^2$ | 20 |
| $\varepsilon$ | $10^{-6}$ |
| $\sigma_\zeta^2$ | 20 |

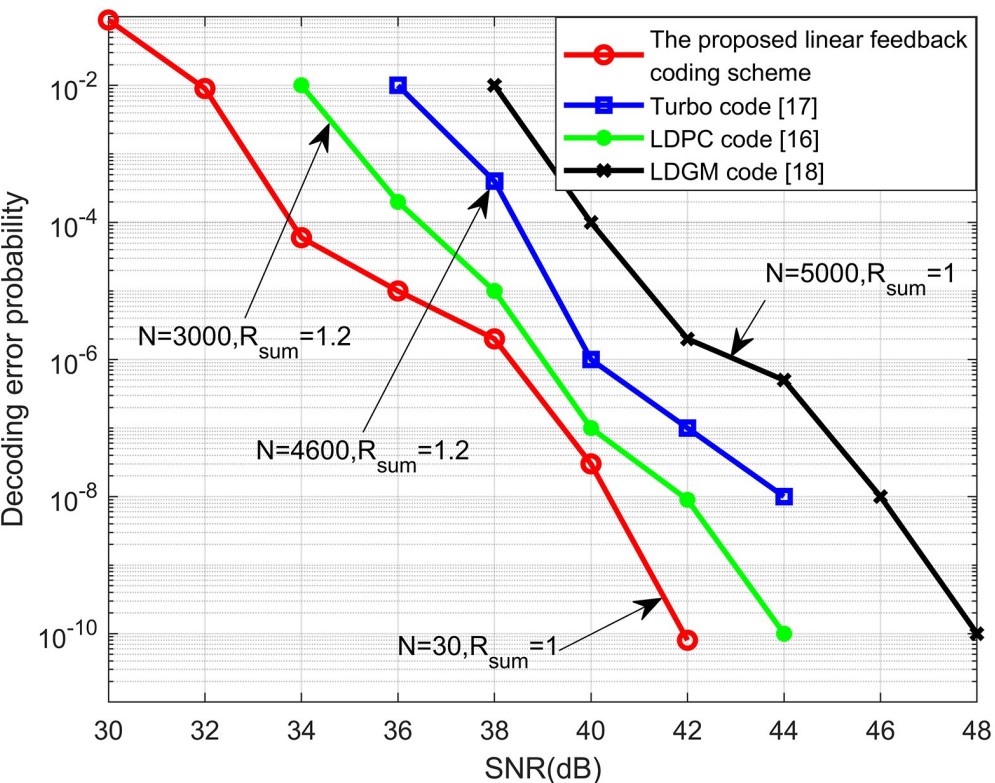

**Fig 6. Comparison of the SNR and decoding error probability of the LDPC code, Turbo code, LDGM code and the proposed linear feedback coding scheme for the FMAC-DMS.**

bit error rate has little effect to the FBL sum rate. Furthermore, the required blocklength for achieving the desired decoding error probability is significantly short. The reason is that the decoding error probability of our scheme doubly exponentially decays as the coding block-length increases.

Fig 10 plots the relationship between the sum rate, decoding error probability and power constraint for the MISO-FMAC-DMS. Note that the power constraint $P_A = P_B = P$, block-length $N = 20$. Furthermore, the LDPC code [15] in Fig 10 based on the secret key scheme [7, 8] (message encrypted by the key). Fig 10 shows that the sum rate is increasing while power constraint and decoding error probability are increasing. Additionally, when the decoding error probability is the same, the sum rate of our scheme is higher than that of LDPC code [15] based on the secret key scheme [7, 8]. This is because the feedback can enlarge the capacity region of the MAC.

Fig 11 shows that when the blocklength is greater than a threshold 129, the achievable secrecy sum rate $R_{sum-miso}(N, \varepsilon, \delta)$ approaches the corresponding achieveable sum rate $R_{sum-miso}(N, \varepsilon)$ for the MISO-FMAC-DMS, which indicates that the proposed scheme is secure when the blocklength is larger than certain threshold. Here note that the secrecy level equal to 1 corresponds to perfect secrecy.

Fig 12 plots the relationship between the blocklength threshold satisfying secrecy con-straint, the decoding error probability, and the secrecy level for the MISO-FMAC-DMS and external eavesdropper. From Fig 12, according to the simulation results, it can be seen that if the decoding error probability is smaller or the blocklength is bigger, the more stringent

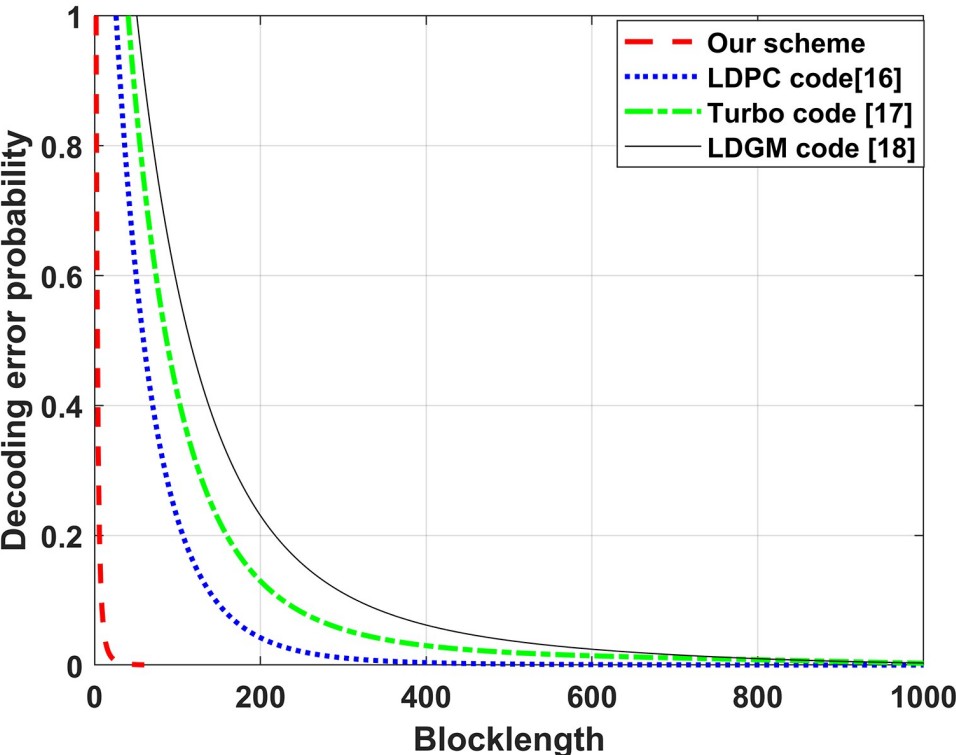

**Fig 7. Comparison of the blocklength and decoding error probability of the LDPC code, Turbo code, LDGM code and the proposed linear feedback coding scheme for the FMAC-DMS.**

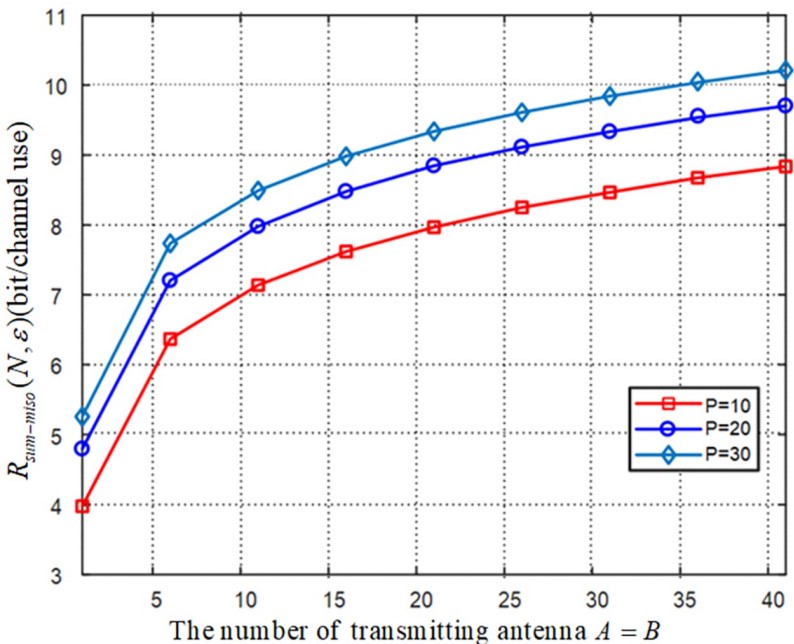

**Fig 8. The relationship between the sum rate, number of transmitting antenna $A = B$ and power constraint $P_A = P_B = P$ for the MISO-FMAC-DMS.**

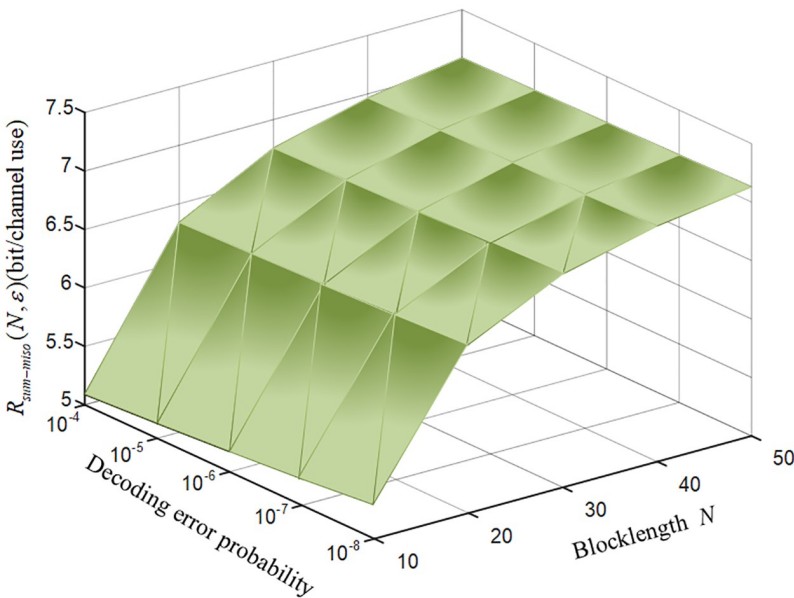

**Fig 9. The relationship between the FBL sum rate, decoding error probability and blocklength for the MISO-FMAC-DMS.**

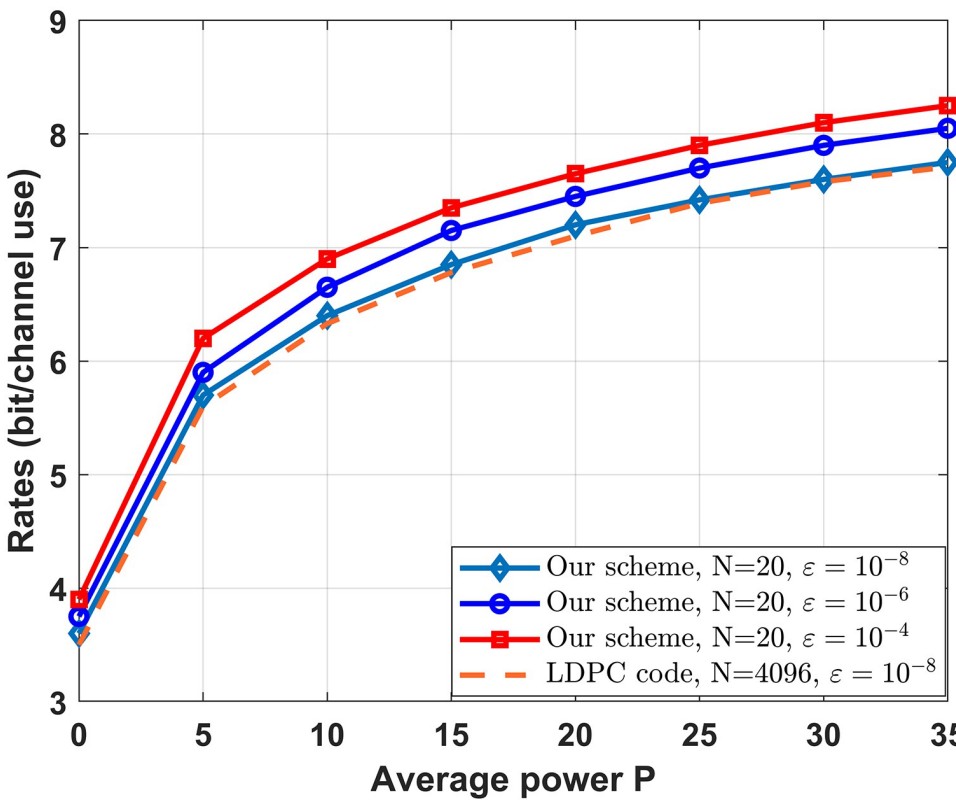

**Fig 10. The relationship between the sum rate, decoding error probability and power constraint for the MISO-FMAC-DMS.**

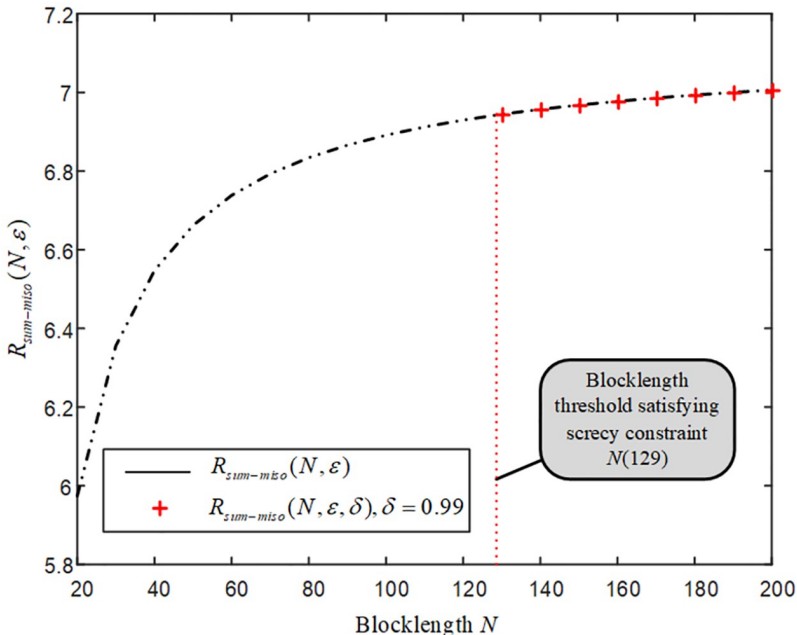

**Fig 11. The sum rate and secrecy sum rate when the blocklength is larger than a threshold for the MISO-FMAC-DMS and external eavesdropper.**

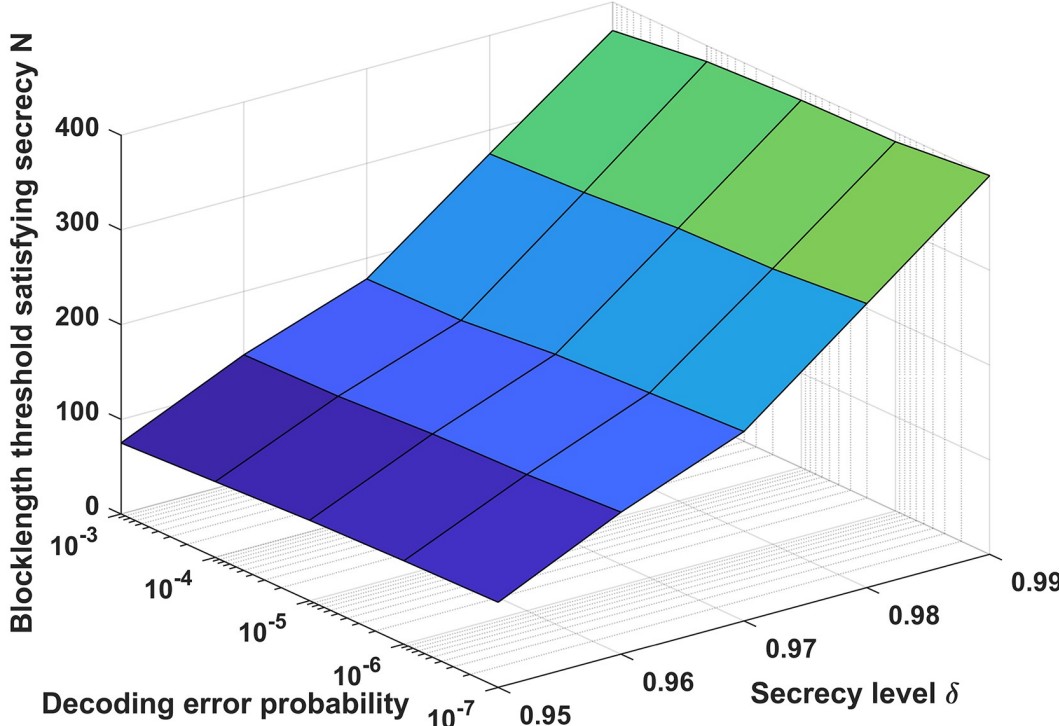

**Fig 12. The relationship between the blocklength threshold satisfying secrecy constraint, the decoding error probability, and the secrecy level for the MISO-FMAC-DMS and external eavesdropper.**

security requirements can be met. Furthermore, the secrecy level increases as the decoding error probability and code blocklength increase, this observation aligns with the theoretical analysis. For example, when the decoding error probability $\epsilon = 10^{-7}$ and the secrecy level $\delta = 0.99$, the blocklength threshold satisfying secrecy constraint is about 400.

## 7 Conclusion and future work

This paper focuses on channel coding from an information theory perspective, addressing the demands for high reliability, low latency, and data transmission security in FMAC-DMS. A linear feedback coding scheme for FMAC-DMS is designed, incorporating the beamforming strategy, channel splitting method, and layered SK coding scheme. The proposed scheme is extended to SISO-FMAC-DMS and MISO-FMAC-DMS. This coding scheme not only provides a FBL coding solution but also guarantees PLS. The subsequent data simulation results demonstrate the effectiveness of the proposed scheme as a channel coding solution, and it ensures PLS when there is an eavesdropper, especially for sufficiently long code lengths. In future work, the application of the proposed scheme to FMAC-DMS with non-causal channel state information (NCSI) at the transmitter can be considered, along with the investigation of FBL achievability and rates, channel capacity, and security capacity in such a system.

## Supporting information

**S1 Appendix.**
(PDF)

## Author Contributions

**Formal analysis:** Yuan Liao.

**Supervision:** Xiaofang Wang.

**Writing – original draft:** Yuan Liao.

**Writing – review & editing:** Xiaofang Wang.

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
