## [Decision Letter · Decision Letter 0]

24 Sep 2023

PONE-D-23-28003Linear Feedback Coding Scheme for Multiple-access Fading Channels with Degraded Message SetsPLOS ONE

Dear Dr. Wang,

Thank you for submitting your manuscript to PLOS ONE. After careful consideration, we feel that it has merit but does not fully meet PLOS ONE’s publication criteria as it currently stands. Therefore, we invite you to submit a revised version of the manuscript that addresses the points raised during the review process.

We look forward to receiving your revised manuscript.

Kind regards,

Praveen Kumar Donta, Ph.D.

Academic Editor

PLOS ONE

Journal Requirements:

Reviewers' comments:

Reviewer's Responses to Questions

**Comments to the Author**

1. Is the manuscript technically sound, and do the data support the conclusions?

Reviewer #1: Partly

Reviewer #2: Yes

Reviewer #3: Yes

2. Has the statistical analysis been performed appropriately and rigorously? 

Reviewer #1: Yes

Reviewer #2: Yes

Reviewer #3: Yes

3. Have the authors made all data underlying the findings in their manuscript fully available?

Reviewer #1: Yes

Reviewer #2: Yes

Reviewer #3: Yes

4. Is the manuscript presented in an intelligible fashion and written in standard English?

Reviewer #1: Yes

Reviewer #2: Yes

Reviewer #3: Yes

5. Review Comments to the Author

Reviewer #1: This manuscript proposed a linear feedback coding scheme for fading multiple-access channels with degraded message sets (FMAC-DMS). My concerns are listed in the following.

1. In Lines 52-56, the introduction of sections is missing.

2. Why there are two repeated “SK feedback scheme for the FMAC-DMS” in Line 108 and 109?

3. In Line 277 and Line 278, please be careful about English writing “From this figure, According to the…”.

4. In figure, you use Fig. But in description, you use Figure. Please unify your expression.

5. “From Figure 6, we see that the required coding blocklength achieving desired decoding error probability is significantly shorter than LDPC, Turbo and LDGM codes.” But I find that after 39 SNR(dB), the LDPC code has shorter error probability than your proposed model. It seems that the trend of plots is different from your conclusion. Could you give some explanations?

6. What are the advantages of your channel coding scheme? What are the contributions of your coding scheme in high reliability, low latency and data transmission security of FMAC-DMS? Please add more experiments to support your conclusions.

Reviewer #2: This presents the main idea of the layered SK scheme and applies it 42

to FMAC-DMS in both single-input single-output (SISO) and multi-input single-output 43

antenna systems (MISO). The channel capacities and FBL achievable rate expressions 44

for both scenarios are derived. The FBL achievable secure rate expressions are also 45

analyzed.

All the objectives are well addressed in a systematic way with a proper english.

Congratulations to the authors for the good work.

Reviewer #3: The authors present the article entitled “Linear Feedback Coding Scheme for Multiple-access Fading Channels with Degraded Message Sets”

The article presents the following general concerns:

Please avoid using first-person sentences. Use instead third or passive-voice sentences.

I suggest to improve the Abstract section. This section should give a pertinent overview of the work by presenting a brief background of the work, methods, results and conclusion. Also, present quantitative results in order to highlight the novelty of the work.

Improve image quality

Validate that all variables and parameters of the equations are correctly defined.

The numbering of the sections that make up the writing needs to be displayed correctly.

Also, I suggest to add a Discussion section. Here, add a table that compares the proposed work vs the already reported in the state-of-the-art in order to highlight the contributions of the work.

Please update your references.

Put a brief introduction between sections and subsections.

6. PLOS authors have the option to publish the peer review history of their article (what does this mean?). If published, this will include your full peer review and any attached files.

Reviewer #1: No

Reviewer #2: No

Reviewer #3: No

---

## [Author Response · Author response to Decision Letter 0]

29 Oct 2023

see the response letter in the attachment.

---

## [Decision Letter · Decision Letter 1]

7 Nov 2023

PONE-D-23-28003R1Linear Feedback Coding Scheme for Multiple-access Fading Channels with Degraded Message SetsPLOS ONE

Dear Dr. Wang,

Thank you for submitting your manuscript to PLOS ONE. After careful consideration, we feel that it has merit but does not fully meet PLOS ONE’s publication criteria as it currently stands. Therefore, we invite you to submit a revised version of the manuscript that addresses the points raised during the review process.

We look forward to receiving your revised manuscript.

Kind regards,

Praveen Kumar Donta, Ph.D.

Academic Editor

PLOS ONE

Journal Requirements:

Additional Editor Comments:

Please address the following before recommend this paper for publication.

1. Summarize all notations used in this paper through a table. Since many notations, summary table helps to cross check when needed.

2. From line 163 (page 6/24) to rest of the document is in italic. please rectify it.

3. Simulation parameters are summarized through a table. It is recommended to provide the source code of simulation through a link.

4. Discussions related to the plots is very limited. Recommended to provide sufficient discussion for each plot describing axis, reasons for improvements of proposed work over others, challenges etc.

Reviewers' comments:

Reviewer's Responses to Questions

**Comments to the Author**

1. If the authors have adequately addressed your comments raised in a previous round of review and you feel that this manuscript is now acceptable for publication, you may indicate that here to bypass the “Comments to the Author” section, enter your conflict of interest statement in the “Confidential to Editor” section, and submit your "Accept" recommendation.

Reviewer #2: All comments have been addressed

Reviewer #3: All comments have been addressed

2. Is the manuscript technically sound, and do the data support the conclusions?

Reviewer #2: Yes

Reviewer #3: Yes

3. Has the statistical analysis been performed appropriately and rigorously? 

Reviewer #2: Yes

Reviewer #3: Yes

4. Have the authors made all data underlying the findings in their manuscript fully available?

Reviewer #2: Yes

Reviewer #3: Yes

5. Is the manuscript presented in an intelligible fashion and written in standard English?

Reviewer #2: Yes

Reviewer #3: Yes

6. Review Comments to the Author

Reviewer #2: The manuscript is well improved by addressing the revision (R1) comments.

Now the paper can be accepted as per the protocol of the journal.

Reviewer #3: (No Response)

7. PLOS authors have the option to publish the peer review history of their article (what does this mean?). If published, this will include your full peer review and any attached files.

Reviewer #2: **Yes: **V. B Murali Krishna

Reviewer #3: No

---

## [Editor Report · Decision Letter 2]

21 Nov 2023

Linear Feedback Coding Scheme for Multiple-access Fading Channels with Degraded Message Sets

PONE-D-23-28003R2

Dear Dr. Wang,

We’re pleased to inform you that your manuscript has been judged scientifically suitable for publication and will be formally accepted for publication once it meets all outstanding technical requirements.

Kind regards,

Praveen Kumar Donta, Ph.D.

Academic Editor

PLOS ONE
---

## [Editor Report · Acceptance letter]

28 Dec 2023

PONE-D-23-28003R2 

PLOS ONE

Dear Dr. Wang, 

I'm pleased to inform you that your manuscript has been deemed suitable for publication in PLOS ONE. Congratulations! Your manuscript is now being handed over to our production team.

Kind regards, 

on behalf of

Dr. Praveen Kumar Donta 

Academic Editor

PLOS ONE